# Using control charts to understand community variation in COVID-19

**Moira Inkelas**[1,2]*, **Cheríe Blair**[3], **Daisuke Furukawa**[3‡], **Vladimir G. Manuel**[2,4‡], **Jason H. Malenfant**[3‡], **Emily Martin**[5‡], **Iheanacho Emeruwa**[2,6‡], **Tony Kuo**[2,4,7,8‡], **Lisa Arangua**[8‡], **Brenda Robles**[8‡], **Lloyd P. Provost**[9]

**1** Department of Health Policy and Management, Fielding School of Public Health, University of California Los Angeles, Los Angeles, California, United States of America, **2** Population Health Program, Clinical and Translational Science Institute, University of California Los Angeles, Los Angeles, California, United States of America, **3** Division of Infectious Diseases, Department of Medicine, David Geffen School of Medicine, University of California Los Angeles, Los Angeles, California, United States of America, **4** Department of Family Medicine, David Geffen School of Medicine, University of California Los Angeles, Los Angeles, California, United States of America, **5** Department of Medicine, David Geffen School of Medicine, University of California Los Angeles, Los Angeles, California, United States of America, **6** Division of Critical Care Pulmonology, Department of Medicine, David Geffen School of Medicine, University of California Los Angeles, Los Angeles, California, United States of America, **7** Department of Epidemiology, Fielding School of Public Health, University of California Los Angeles, Los Angeles, California, United States of America, **8** Division of Chronic Disease and Injury Prevention, Los Angeles County Department of Public Health, Los Angeles, California, United States of America, **9** Associates for Process Improvement, Austin, Texas, United States of America

☯ These authors contributed equally to this work.
‡ These authors also contributed equally to this work.
* minkelas@ucla.edu

**Data Availability Statement:** All relevant data are uploaded to GitHub (https://github.com/datadesk/california-coronavirus-data#latimes-county-totalscsv).

## Abstract

Decision-makers need signals for action as the coronavirus disease 2019 (COVID-19) pandemic progresses. Our aim was to demonstrate a novel use of statistical process control to provide timely and interpretable displays of COVID-19 data that inform local mitigation and containment strategies. Healthcare and other industries use statistical process control to study variation and disaggregate data for purposes of understanding behavior of processes and systems and intervening on them. We developed control charts at the county and city/neighborhood level within one state (California) to illustrate their potential value for decision-makers. We found that COVID-19 rates vary by region and subregion, with periods of exponential and non-exponential growth and decline. Such disaggregation provides granularity that decision-makers can use to respond to the pandemic. The annotated time series presentation connects events and policies with observed data that may help mobilize and direct the actions of residents and other stakeholders. Policy-makers and communities require access to relevant, accurate data to respond to the evolving COVID-19 pandemic. Control charts could prove valuable given their potential ease of use and interpretability in real-time decision-making and for communication about the pandemic at a meaningful level for communities.

**Funding:** The research described was supported, in part, by the National Institutes of Health /National Center for Advancing Translational Sciences through UCLA CTSI Grant Number UL1TR000124 and the UCLA CTSI TL1 Grant Number TL1TR001883. The content is solely the responsibility of the authors and does not necessarily represent the official views of the National Institutes of Health. API provided support in the form of salary for author LP but did not have any additional role in the study design, data collection and analysis, decision to publish, or preparation of the manuscript. The specific roles are articulated in the 'author contributions' section.

**Competing interests:** API provided support in the form of salaries for author LP. This does not alter our adherence to PLOS ONE policies on sharing data and materials.

# Introduction

Coronavirus disease 2019 (COVID-19) mitigation and containment policies have significant economic, social, and health impact. Enacting sensible public policies in the COVID-19 pandemic requires real-time data that public leaders can easily interpret and act on. The constituencies for these data are expanding as regional and community stakeholders, including cities, businesses, and school districts, assume decision-making roles in the emergency response. Moreover, given that public health interventions require public cooperation and trust in guidance and decisions that are data-driven, there is also a need to engage the public at large to successfully implement mitigation and containment strategies. Offering data to inform policy and individual health behavior is a cornerstone of prevention and public health practice [1].

While providing relevant, accessible, and timely data is a core public health function, current displays of COVID-19 data lack features that policy-makers require for decision-making and that communities need to make the connection between their actions and the state of the pandemic. Such displays often use maps, day-to-day percentage changes, and cumulative counts that refresh daily. These formats obscure variation across places, populations, and time, which are essential to learning how actions and events affect COVID-19 cases and deaths. Over-aggregation impairs the ability of decision-makers to make real-time policy adjustments and to assess the impact of these adjustments. The public is unable to see local data that are most relevant and motivating to them regarding health behaviors [2].

Statistical process control method and theory focuses on ease of use and interpretation for end users [3–7] and learning under conditions of uncertainty [8, 9]. Many commercial, healthcare, and education organizations use control charts to understand the behavior of processes or systems over time [8–11]. By distinguishing random ("common cause") variation from non-random ("special cause") variation, control charts reduce over-reaction to noise in data while enabling timely response when true signals show that conditions are improving or deteriorating [11]. They enable scientists, policy-makers, and community members alike to learn if a change to a policy or process has affected an outcome of interest [11–13]. Control charts can be used for multiple common types of data distributions including classification (binomial) P charts, continuous (Xbar charts) and individuals (I charts), and count (Poisson) C charts or U charts [12].

Despite their potential value, control charts are not part of standard public health practice [11–16]. This article illustrates how control charts can be used to achieve public health goals in the COVID-19 pandemic. We offer prototype control charts and displays to demonstrate their utility.

# Methods

## Statistical process control

Control charts display data in an ordered format, most often ordered over time, to understand, manage, and improve the behavior of a specific process or system. The control chart includes a centerline (i.e., the mean of the data) and upper- and lower-control limits, which are three sigma above and below the centerline. When the measure is stable over time, the centerline and limits provide a rational prediction of future observations [12]. Values outside of the control limits indicate that the outcome is not being produced from one consistent homogeneous process [3, 4]. When there is a signal of change, the centerline and control limits shift to reflect the new level of performance [12]. This study uses a hybrid control chart for count data and exponential growth or decline (I chart) developed by Perla et al. for use in a pandemic [17–19].

## Data sources on COVID-19 cases

This study analyzed data for selected regions of California. The county-level control charts use daily counts of COVID-19 from the Los Angeles Times COVID-19 repository, which provides a public datafile of cases reported by California counties [20]. The original data source is the Confidentiality Morbidity Report (CMR16) of laboratory-confirmed COVID-19 that counties report to the California Department of Health Care Services. The Los Angeles County Department of Public Health (LAC DPH) reports data for 272 distinct cities and neighborhoods.

## Control chart analysis

The control charts display daily reported COVID-19 cases. Charts for counties begin on March 2, 2020 and charts within LA County begin on March 16 for consistency. This study uses the hybrid control chart method developed by Perla et al. [17] to view epochs and phases in the pandemic. The four possible epochs are pre-exponential growth (C-chart), exponential growth (an individuals (I) chart fitted to log10 of the data series and transformed back to the original scale), post-exponential growth (a flat trajectory or exponential decline that is represented by an I chart), and stability after descent (C-chart) [17, 21, 22]. A region may experience one or multiple epochs. A phase is a time period that is represented by a distinct control chart; there can be multiple phases within an epoch.

To estimate the centerline and upper and lower control limits, the method requires at least eight observations to meet the minimum requirements for an effective C-chart [12] in the first and fourth epochs. The control charts automatically set the limits of the exponential growth period based on regression analysis of the first 20 observations [17]. The exponential growth phase is modeled by the log-linear regression I-chart [12]. We used model coding in R developed by Perla et al. [21, 22] to transform the counts using the log10 function and calculated the intercept and slope through regression analysis for the log10 data; the regression line becomes the centerline (CL). Limits for the exponential phase in the charts are calculated from the median moving range of the residuals, with the upper limit (UL) and lower limit (LL) calculated as CL+3.14*MRbar and CL−3.14*MRbar, respectively. The CL, UL, and LL were then transformed to the original count scale. Charts that do not display an exponential growth phase are in C-chart format for the full period studied.

Formal use of control charts identifies special cause through established statistical rules combined with inspection by experts in the system being studied. Standard criteria for special cause are an observation outside of an upper or lower control limit or a shift of 8 successive observations above or below the centerline [11, 12]. For the exponential Epochs 2 and 3, we used Shewhart criteria modified by Perla et al. [17, 22], which require two points rather than one point above the control limits to signal the start of a new phase. The rationale is that COVID-19 data displays more than "usual" variation in the form of single large values that reflect "data dumps" from reporting entities; requiring a stronger signal prevents such a data artifact from triggering a new phase [22].

Notably, time series charts often reveal reporting artifacts. An example is peak values early in the week due to cases accumulating over a weekend. It is common practice in control charts to remove special cause variation due to such cyclic behavior by separating data lines within a chart or by subgrouping by a larger unit (i.e. week rather than day) to smooth variation. In this study, we preserved daily periodicity based on the statistical principles underlying control chart methodology, which is that data should not be summarized if it would mislead the user into taking actions that would not be taken had the original data been preserved [6]. Smoothing the data in these COVID-19 control charts would temper but not remove the apparent case reporting artifact, and seeing these patterns offers insights, namely that there is an impact from facility case reporting.

## Description of California counties and LA County subregions included in the analysis

California is home to more than 12 percent of the U.S. population and has a complex geopolitical landscape with 58 distinct counties. LA County is a vast region with over 10 million residents, 88 cities (including the City of Los Angeles), 272 designated neighborhoods, and three public health departments, of which the largest is the LAC DPH.

**Selection of counties and LA County subregions for inclusion in the analysis.** The study team selected an illustrative set of charts using criteria relevant to the COVID-19 pandemic. We included five counties from the second, third, and fourth quartiles for population and from northern, central, and southern regions of California. We selected one neighborhood and four cities within LA County. Within LA County, we sought variation in sociodemographic factors that we considered to be especially relevant to the COVID-19 pandemic: median income, overall health, median age, race/ethnicity, population density (people per square mile), median household size, and percentage of households that experience household crowding, which is a measure derived from the U.S. Census that is defined as the percentage of households with a ratio of total household members to rooms (excluding bathrooms) greater than one.

**Data sources.** Table 1 shows sociodemographics of selected areas. Measures of population, race/ethnicity, median income, household size, and population density come from the United States Census Bureau QuickFacts (2019) [23]. Median age comes from the 2018 American Community Survey (ACS) published by Towncharts [24]. Household crowding comes from the California Healthy Places Index (Public Health Alliance of Southern California) based on an ACS five year average for 2011–2015 [25]. Quartiles of health come from an overall ranking developed by County Health Rankings & Roadmaps (University of Wisconsin) that combines multiple health outcomes including premature death, poor or fair health, poor physical health days, poor mental health days, and low birthweight [26]. For the neighborhood within LA County, race/ethnicity comes from L.A. Mapping (Los Angeles Times) [27].

Table 1 also shows the proportion of COVID-19 cases from congregate living facilities in the first four months of the pandemic (March through June 2020); these data are available for some but not all counties through their public health department websites. For LA County overall and for cities and neighborhoods within the County, the congregate counts include cases from residential health and living facilities including skilled nursing facilities (SNFs), shelters, and correctional facilities. The congregate measure for Santa Clara County includes only residential long term care facilities [28]. The LA County DPH website provides COVID-19 test volume rates per 100,000 population by area, based on electronic lab reporting; the number per 100,000 in January 2021 showed modest variation for the areas in this study: 32,569 for Santa Monica, 19,279 for Lancaster, 26,980 for Bell, and 26,730 for Westlake.

## Results

Figs 1 and 2 show control charts for five counties. Four counties experienced exponential growth in COVID-19 cases in early March 2020, which was followed by a period of lower exponential growth in three and non-exponential growth in one. Each county experienced at least one period of exponential growth; one county (Santa Clara) experienced no exponential growth until a November surge that was observed in all counties. Imperial County showed a cyclic weekly pattern associated with the lack of case reporting on weekends; this analysis retained all days in the chart for comparability with other counties.

There was considerable variation in COVID-19 cases over time among the studied subregions in LA County. Two (Lancaster, Fig 1 and Lynwood, Fig 2) experienced initial

**Table 1. Characteristics of selected California counties and neighborhood/cities within Los Angeles County.**

| Region | Population | Health quartile | Median age | Race/ethnicity | Median income | Population density | Median household size | % crowded households | % congregate cases as of 6/30/20 |
|---|---|---|---|---|---|---|---|---|---|
| **Counties** | | | | | | | | | |
| Los Angeles | 10,039,107 (1, 4th) | 3rd | 36 | 49% Latino, 9% Black, 15% Asian | $64,251 | 2,420 | 3.0 | 14 | 15 |
| San Diego | 3,338,330 (2, 4th) | 4th | 36 | 34% Latino, 6% Black, 13% Asian | $74,855 | 736 | 2.9 | 7 | --[a] |
| Santa Clara | 1,922,852 (6, 4th) | 4th | 37 | 25% Latino, 3% Black, 38% Asian | $116,178 | 1,381 | 3.0 | 8 | 13[b] |
| Solano | 447,643 (20, 3rd) | 3rd | 38 | 27% Latino, 15% Black, 16% Asian | $77,609 | 503 | 2.9 | 5 | --[a] |
| Imperial | 181,215 (30, 2nd) | 1st | 32 | 85% Latino, 3% Black, 2% Asian | $45,834 | 42 | 3.9 | 10 | --[a] |
| **Cities/Neighborhood** | | | | | | | | | |
| Lancaster | 157,601 | 1st | 32 | 40% Latino, 22% Black, 4% Asian | $52,504 | 1,661 | 3.2 | 4 | 26 |
| Westlake | 103,839 | 1st | 27 | 73% Latino, 4% Black, 16% Asian | $26,757 | 38,214 | 3.0 | 45 | 25 |
| Santa Monica | 84,084 | 4th | 38 | 16% Latino, 4% Black, 10% Asian | $93,865 | 10,664 | 2.0 | 2 | 46 |
| Lynwood | 71,022 | 1st | 30 | 88% Latino, 9% Black, 1% Asian | $49,684 | 14,416 | 4.4 | 33 | 28 |
| Bell | 36,667 | 1st | 24 | 92% Latino, 2% Black, 1% Asian | $42,548 | 14,185 | 4.0 | 27 | 15 |

Data sources: Demographics from the United States Census Bureau QuickFacts, County Health Rankings (University of Wisconsin Population Health Institute), L.A. Mapping (Los Angeles Times). Congregate cases from the Los Angeles County Department of Public Health COVID-19 Dashboard, California county COVID-19 websites. Accessed June 30, 2020.

[a]Indicates that data are not publicly available.

[b]Includes only congregate health and living facilities (not correctional facilities).

exponential growth followed by a second phase of exponential growth with a lower midline before transitioning to a non-exponential epoch. Another (Westlake) experienced initial exponential growth and then entered a non-exponential epoch with multiple phases through the study period. This neighborhood had the highest residential density and household overcrowding relative to others as well as large household size (median of 3.0) (Table 1). Other areas with relatively high crowding and household size experienced multiple phases but no exponential rise until late 2020. The city with the highest median income and lowest rate of overcrowding (Santa Monica) showed low case counts throughout the study period with a doubled rate in the last month of the study. For the first four months of the study period, nearly half (46%) of cases in this city came from congregate facilities, while the rate ranged from 15% to 28% for other areas.

Several subregions show one or two non-sequential daily rates that exceed the upper control limit; these may be due to reporting patterns from laboratories or the public health department.

Fig 2 shows charts for the City of Lynwood and LA County with annotated events such as public health authority COVID-19 orders, introduction of free testing centers, and holidays. The Lynwood chart is annotated with additional policies that are specific to this city, such as mandated use of facial coverings in public in the first week of April, about six weeks earlier than LA County. The charts show different time trends. The exponential epoch ended in

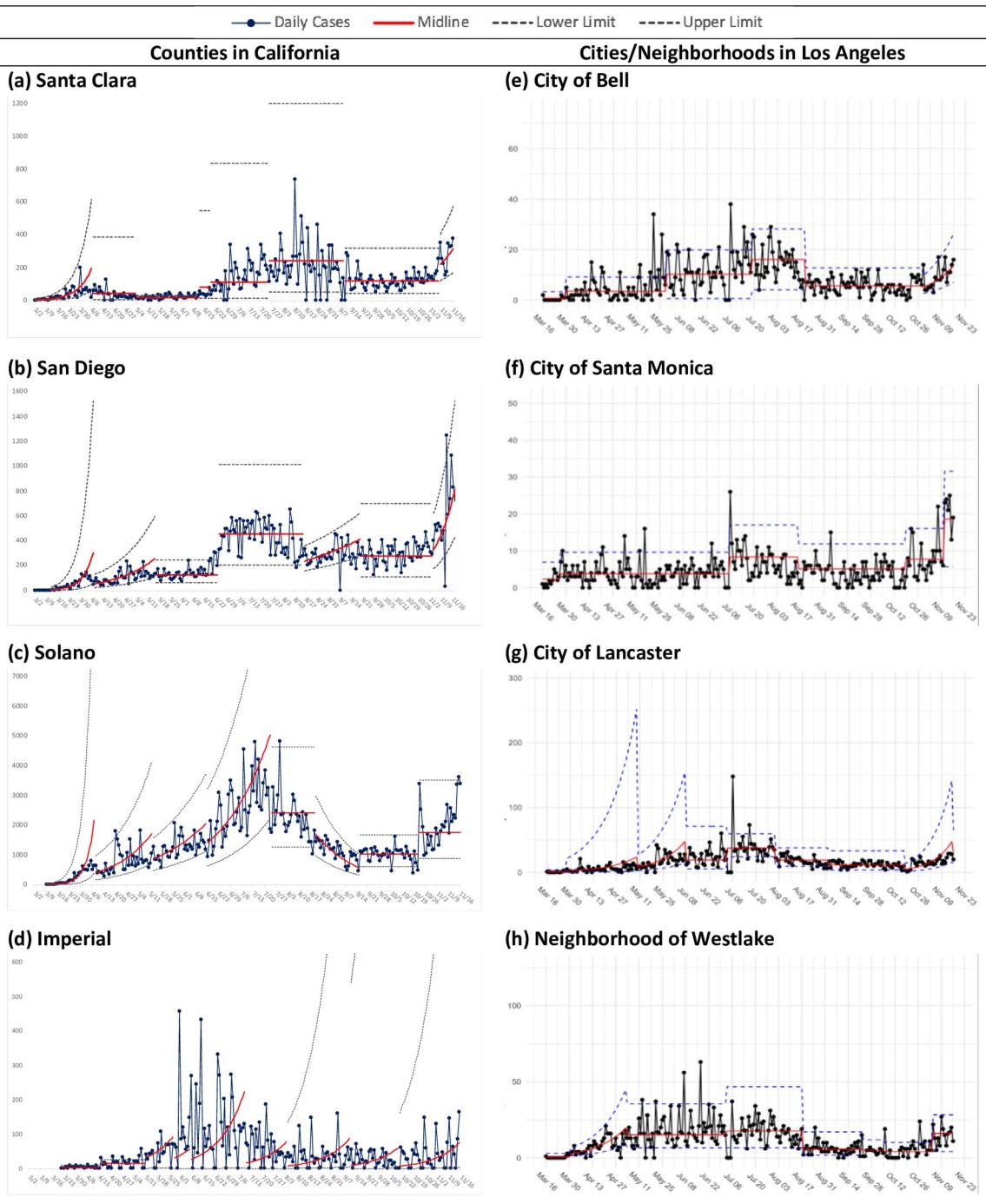

**Fig 1. Control charts of COVI D-19 cases: California, counties, subregions.** Shows daily case counts, midline, and upper and lower control limits. Source for county data is the New York Times. Source for Los Angeles cities/neighborhoods is the Department of Public Health COVID-19 dashboard (accessed 1/10/2020).

Lynwood in June; a new phase of exponential rate in LA County began at the end of that month. Neither chart shows special cause several weeks after major holidays including Mother's Day, the Fourth of July, and Labor Day.

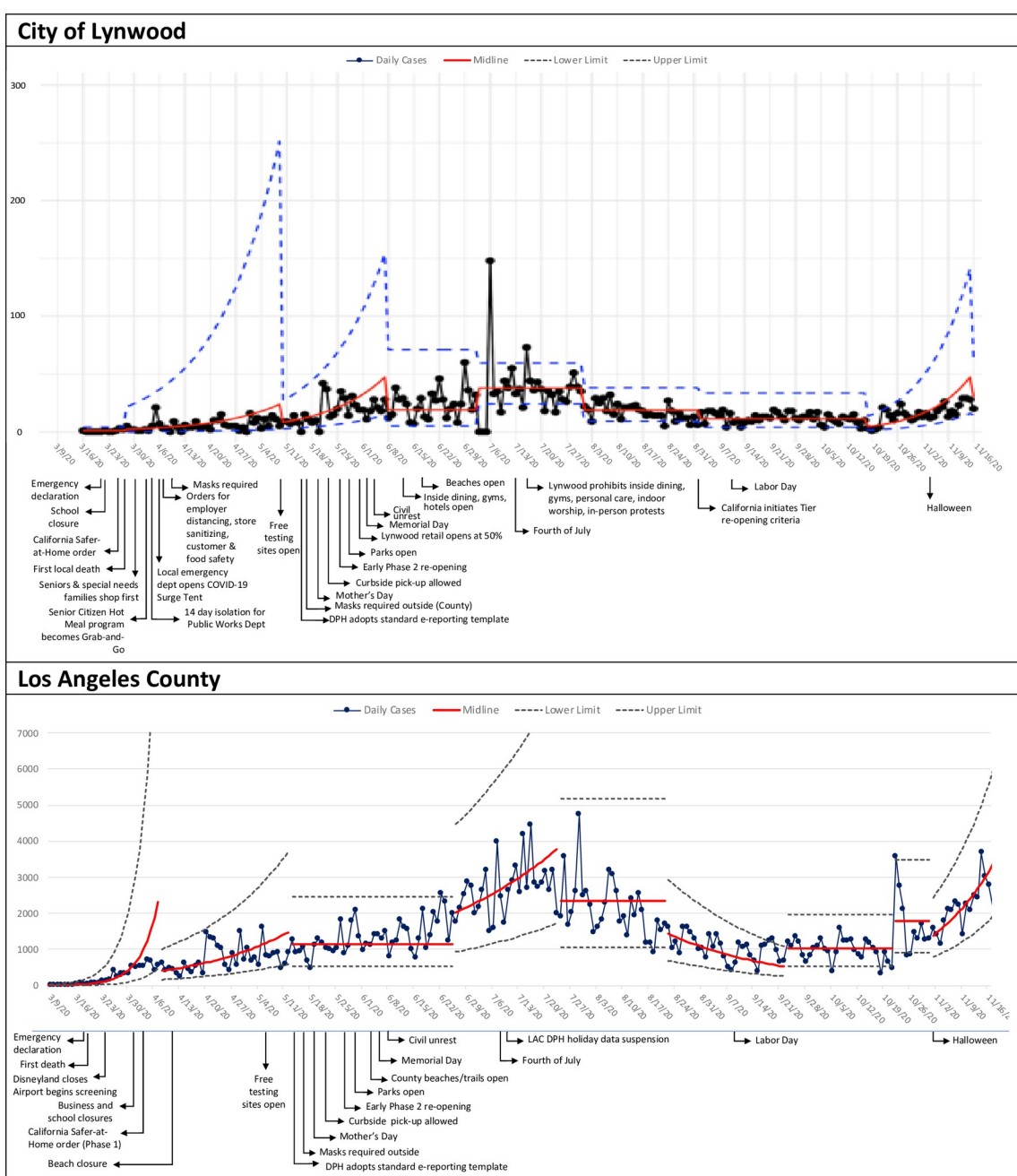

**Fig 2. Annotated control charts of COVID-19 cases: City of Lynwood and Los Angeles County.**

## Discussion

During the COVID-19 pandemic, decision-makers need signals from data to intensify or relax safety requirements. Control charts are a tool for daily learning and action that public health could benefit from [29–31]. This study shows how a novel hybrid control chart could assist authorities to act early by signaling exponential and non-exponential growth or decline in public health measures. Control charts are commonly used in management and can be used in public health to distinguish random from meaningful variation. Use of this method could

reduce overreaction to expected random variation and encourage immediate action when special cause variation signals a new phase or epoch of the pandemic.

Based on this study, a division within LAC DPH incorporated control charts into their approach for identifying COVID-19 outbreaks.

This study shows the value of disaggregating data as the large-scale closures that characterized the initial emergency response to COVID-19 evolved into metric-focused decision-making. California adopted six indicators for relaxing the initial safer-at-home order [32] and introduced a tiered system based on local data in October 2020 [33] as local authorities adopted additional policies. Disaggregation by region and time helps state, county, and within-county authorities act on the data. For example, local authorities could use the significant within-county variation to communicate to the public what is happening in specific neighborhoods and potentially to consider mitigation strategies that customize to subregions. Further, the study shows that a city could be mischaracterized as experiencing an outbreak of COVID-19 cases because case counts for the general public and congregate living populations are combined. This was especially relevant early in the pandemic when skilled nursing facilities accounted for a significant proportion of COVID-19 cases. Local jurisdictions may need this differentiation to target strategies–for example, if most cases are in congregate facilities. Disaggregated data could also enable a school district to consider if patterns in neighborhoods that feed into specific schools warrant augmented mitigation strategies.

Control charts such as those in this study can be easily interpreted once their format is understood. They can help localities check assumptions about time trends and assess the impact of planned changes (policies) and external influences (such as holidays). Their visual simplicity is designed for analysis, discussion, and decision-making. The disaggregated, time-ordered format can empower people in a specific city or neighborhood to identify potential causes of what they observe in the data. These COVID-19 charts may provide the same benefit to local communities during the pandemic as control charts offer to industries that use them for improving clinical care or other processes.

The interpretation of these charts is aided by annotation of key events. Especially when done in real time, annotation makes it possible to observe if specific policies or events are followed by signals of a new phase or epoch. This study shows that assumptions about increased COVID-19 cases following holidays such as Mother's Day, the Fourth of July, and Labor Day were not borne out in the county or neighborhood data. Users of these control charts can employ the Bradford Hill causality criteria to aid interpretation [34]. Additionally, complementing control charts with a small number of meaningful measures, such as residential density and household size, helps decision-makers and the public interpret observed variation by area and over time. Local health agencies could work with cities to identify appropriate measures that aid interpretation, such as housing features and rates of mask-wearing. Other analytical methods such as modeling associations of demographics or other subregion features with case counts can complement control charts.

Lastly, the time-ordered format of a control chart contrasts with how public health data are often analyzed and displayed. Reporting patterns by public health departments (e.g., batch reporting by facilities, weekday versus weekend reporting, inclusion of congregate living residents and homeless individuals in cumulative case reporting, availability of illness dates versus reporting dates) could shape how stakeholders interpret the data. At minimum, control charts can make these limitations transparent and prompt decision-makers to ask for more interpretable displays. Engaging decision-makers in such discussions may help to justify public health investment in data reporting and analytics so that such questions can be answered from the outset in future outbreaks.

Notably, this study is limited in the interpretation of the displayed control charts. Statistical process control was intended for use in real time by people with intimate knowledge of the system, process, or community that the underlying data represent. Proper interpretation of data comes from engaged discussion. This study offers some possible interpretations of the observed special cause variation, but the purpose was to illustrate the approach rather than draw causal conclusions about policy events and COVID-19 cases. Rather than interpreting the data for action, the study intended to examine if areas exhibited variation and to illustrate the potential value of providing disaggregated data in control chart format to people who can make immediate use of them. Finally, COVID-19 case counts underestimate total COVID-19 cases, and there may be noise in the case counts that result from limited staffing for data entry, delayed reporting of cases out of jurisdiction, and the impact of time-varying testing constraints and laboratory turn-around-time, among others, which may vary by county and over time.

As with all significant changes to data reporting and analytics, adopting control chart methods will require time and resource investment. LAC DPH's interest in using control charts to help inform outbreak management followed a series of virtual workshops on the method that a university partner offered to public health personnel. Health departments will need to create these displays and prepare their workforce to use them effectively, ideally as real-time learning tools. For optimal impact, public health personnel will need to assist community stakeholders to interpret and use the charts to meet their specific needs. Progress in this area for health authorities and their partnered community stakeholders has been made for topics such as infant mortality through learning collaboratives of state and local health departments [14] and in training programs for local health authorities [31].

## Conclusions

The COVID-19 pandemic has placed unprecedented data and analytic demands on public health. Clear, interpretable, disaggregated displays are essential tools for policy makers as they consider the health, economic, social, mental health, and educational burden of COVID-19 in their communities. Showing the data underlying public policy decisions could increase uptake of guidance regarding personal and collective behavior in communities. This may have been a missed opportunity as the COVID-19 pandemic progressed from an initial, centralized emergency response to a period of distributed decision-making on the part of employers, cities, school districts, and others. Healthcare organizations use control charts for learning and to encourage data-driven collective action [15, 35, 36]. Public health and population-oriented systems could also make use of this method. This statistical method provides an easy, effective, and inexpensive way for public health departments to meet some of the informational needs of governors, city councils, school boards, and the general public during an emergency response when timeliness of and insight from data are of utmost importance. Use of control charts in public health will require an appreciation of their value and investment of time and resources into their creation and use.

## Author Contributions

**Conceptualization:** Moira Inkelas, Vladimir G. Manuel, Jason H. Malenfant, Tony Kuo, Lloyd P. Provost.

**Data curation:** Cheríe Blair, Daisuke Furukawa, Emily Martin, Lisa Arangua, Lloyd P. Provost.

**Formal analysis:** Moira Inkelas, Daisuke Furukawa, Jason H. Malenfant, Lloyd P. Provost.

**Methodology:** Moira Inkelas, Lloyd P. Provost.

**Visualization:** Lloyd P. Provost.

**Writing – original draft:** Moira Inkelas, Vladimir G. Manuel.

**Writing – review & editing:** Cheríe Blair, Daisuke Furukawa, Emily Martin, Iheanacho Emeruwa, Lisa Arangua, Brenda Robles, Lloyd P. Provost.

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
