## [Decision Letter · Decision Letter 0]

9 Dec 2020

PONE-D-20-24209

Using Control Charts to Understand Community Variation in COVID-19

PLOS ONE

Dear Dr. Inkelas,

Thank you for submitting your manuscript to PLOS ONE. After careful consideration, we feel that it has merit but does not fully meet PLOS ONE’s publication criteria as it currently stands. Therefore, we invite you to submit a revised version of the manuscript that addresses the points raised during the review process.

The manuscript has been evaluated by two reviewers, and their comments are available below.

The reviewers have raised a number of major concerns. They request improvements to the introduction and justification for use of control charts, reporting of methodological aspects of the study, and further discussion of your findings or alternative approaches.

Could you please carefully revise the manuscript to address all comments raised?

We look forward to receiving your revised manuscript.

Kind regards,

Beryne Odeny

Staff Editor

PLOS ONE

Journal Requirements:

2)  Thank you for stating the following in the Financial Disclosure section:

[National Institutes of Health

/National Center for Advancing Translational Sciences Grant Number UL1TR000124

and Grant Number TL1TR001883. The funders had no role in study design, data

collection and analysis, decision to publish, or preparation of the manuscript.].   

We note that one or more of the authors are employed by a commercial company: Associates for Process Improvement,

i. Please provide an amended Funding Statement declaring this commercial affiliation, as well as a statement regarding the Role of Funders in your study. If the funding organization did not play a role in the study design, data collection and analysis, decision to publish, or preparation of the manuscript and only provided financial support in the form of authors' salaries and/or research materials, please review your statements relating to the author contributions, and ensure you have specifically and accurately indicated the role(s) that these authors had in your study. You can update author roles in the Author Contributions section of the online submission form.

ii. Please also provide an updated Competing Interests Statement declaring this commercial affiliation along with any other relevant declarations relating to employment, consultancy, patents, products in development, or marketed products, etc. 

Reviewers' comments:

Reviewer's Responses to Questions

**Comments to the Author**

1. Is the manuscript technically sound, and do the data support the conclusions?

Reviewer #1: Yes

Reviewer #2: Yes

2. Has the statistical analysis been performed appropriately and rigorously? 

Reviewer #1: Yes

Reviewer #2: Yes

3. Have the authors made all data underlying the findings in their manuscript fully available?

Reviewer #1: Yes

Reviewer #2: Yes

4. Is the manuscript presented in an intelligible fashion and written in standard English?

Reviewer #1: Yes

Reviewer #2: Yes

5. Review Comments to the Author

Reviewer #1: Review PONE-D-20-24209

This is a well written and interesting article illustrating how improved presentation and visualization of data can improve learning and decision-making related to the ongoing Covid pandemic. I have some suggestions that authors may consider when revising and finalizing their manuscript.

The title of the article as well as the abstract focus on the use of control charts, i.e. how data is displayed. I would say that the first strong argument being made in the article is the use of disaggregate data, in particular as more localized decisions are needed to handle existing variations in the pandemic. That data should be presented in the form of control charts, to handle randomness and identify significant changes, is the second argument. I miss explicit references to disaggregated data in the abstract (and perhaps in title). I also miss references to randomness to explain the value of control charts in the abstract (especially when disaggregate data is used, i.e. fewer observations). Also in the introduction, the focus is on “displays of Covid-19 data”. The problem with over-aggregation is mentioned (impairing the ability of decision-makers) but I think this could be presented as a (first) separate issue.

On page 4 it is argued that control charts “are less burdensome” and on page 12 you say that “the analysis show that control charts are easily interpretable” and “have a visual simplicity”. First of all you have not analyzed if control charts are easily interpretable, at least not from a (real) decision-maker point of view. There is no data based on decision-makers views. The views expressed are rather arguments usually put forward by scholars, proposing control charts, funnel plots and similar displays to identify special cause variation. If think (and this is supported by experiments) that real decision-makers need at least initial support to fully understand control charts. Support becomes even more important when decision-making is distributed across local authorities (with more limited access to “experts”). Without initial and continued support (training seminars/webbinars, on-line help etc) it would probably be difficult to develop this into “a community engagement tool”. It would also take time - the Covid pandemic require local actions here and now! In summary, I think the article should take the need for training and support involving local decision-makers more seriously. Continued support could also include sharing of local experiences and local knowledge. This could support engaged discussions and improved decisions further.

Reviewer #2: The authors developed a unique system of C and I-type control charts in order to retrospectively model the first 4 months of the COVID-19 pandemic within California counties and cities/neighborhoods. These charts detected both periods of exponential growth and periods of stable rates for the counties and cities/neighborhoods analyzed. The authors showed how annotated control charts could provide near real-time data to decision makers and the general population, as well as how these charts might provide hypotheses to explain special cause variation.

The manuscript is interesting and well written. As the authors point out, use of control charts to monitor healthcare processes has become more common, but use within the public health arena remains surprisingly uncommon to date.

I do have several comments and suggestions, primarily targeting the methods, that if addressed might improve the readability and impact of this manuscript:

1) Methods. In the first paragraph of the methods, the authors describe the characteristics of the control charts that they used in this analysis. However, these specific characteristics do not apply to all control charts relevant to this type of analysis, and I think this introductory paragraph to control charts should emphasize the wide variation in chart characteristics. Perhaps this paragraph would also be better suited for the introduction rather than the methods?

For example, some control charts estimate the centerline from the mean of past data, but other control charts use an expected external baseline rate or other calculations to estimate the baseline. Furthermore, “freezing” a static baseline is one strategy, but other control charts use a dynamic rolling baseline that updates over time. Finally, defining control limits at +/- 3 standard deviations is a common convention but is not required.

2) Methods – control chart analysis. Exponential outbreak growth was quite common in many locations at the onset of the COVID-19 pandemic. The C-charts in this study estimated a centerline from the first 20 observations. However, as expected for a respiratory virus outbreak, exponential growth often occurred within the first 20 days/data points. Does use of a centerline estimated from the first 20 days of data decrease the ability of these charts to demonstrate timely detection of exponential growth within the first 20 days? Does the 20-day baseline need to be separately analyzed for exponential growth before being used as an “in control” baseline for a process assumed not yet to demonstrate exponential growth? I.e., would the baseline be more effective if it were proven to be “in control” before being used to monitor for special cause variation?

3) Methods/Figure 1.

a) Charts c/d/e/f were deemed not to exhibit exponential growth, but many data points ultimately exceeded the apparent upper control limits for these charts. How were these data indicating special cause variation and possible transition to an exponential growth phase determined not to represent exponential growth?

b) Chart g is labeled as both a hybrid chart and a C-chart – could the authors please clarify this unique feature of the Lancaster control chart?

c) The Figure 1 legend states that county C-charts used a centerline estimated from the midpoint of all observations, but the methods state that C-chart centerlines were estimated from the first 20 data points. Could the authors please reconcile this discrepancy? Also, does “all observations” refer to all observations up to the current observation or all observations for the entire study? I am asking this question because if observations from May were used to calculate the centerline for April, this method of using a global baseline would not be useful for prospective surveillance.

d) Could the authors please label Figure 1? Perhaps a single label for all panels would work well, but the “daily counts,” centerline, UCL, and LCL should be labeled, as in Figure 2.

4) Methods – control chart analysis. The authors defined the end of the exponential phase as when a data point is below the lower limit – when this happens, should an I-Chart be converted back to a C-chart to monitor for re-entry into an exponential phase? If not, it seems that ongoing use of the exponential centerline and control limits is not very useful.

5) Methods – control chart analysis. My comment here is a generalization of comment #4 above. At the bottom of page 6, the authors state that the centerline and control limits remained constant over time, except when an initial transition to epidemic growth was noted. I think this strategy is a major limitation of this analysis that should at least be discussed in more detail or, ideally, addressed by showing how centerlines and control limits could be adjusted overtime as proof of concept for using these charts for prospective surveillance. For example, in Figure 1h, the majority of datapoints from mid-May onwards are above the apparent UCL, and every data point over this broad timeframe is above the centerline. Clearly, a mechanism is needed to update the centerline and control limits over time for COVID-19 monitoring in order for this chart type to remain useful outside of monitoring for exponential growth during the early stage of the outbreak.

6) Results – page 14, just prior to conclusions. The authors state that “the purpose was to draw conclusions about causal events,” but I believe this represents a typo and was intended to state that “the purpose was NOT to draw conclusions…”

7) Overall comment – did the authors explore other chart types and chart characteristics for this analysis? How would other charts perform in certain scenarios for monitoring COVID-19 trends? For example, did the authors experiment with different centerline definitions, out of control data point definitions, control limits, etc.? Per my comment above, would other charts provide more useful monitoring after exponential growth has been confirmed or for locations that continue to see variation in case rates without exponential growth?

6. PLOS authors have the option to publish the peer review history of their article (what does this mean?). If published, this will include your full peer review and any attached files.

Reviewer #1: No

Reviewer #2: No

---

## [Author Response · Author response to Decision Letter 0]

10 Feb 2021

Thank you for the helpful reviews. We agree with and have addressed each comment in the revised narrative. Responses to the reviews are provided below.

Reviewer #1: Review PONE-D-20-24209

I would say that the first strong argument being made in the article is the use of disaggregate data, in particular as more localized decisions are needed to handle existing variations in the pandemic. That data should be presented in the form of control charts, to handle randomness and identify significant changes, is the second argument. I miss explicit references to disaggregated data in the abstract (and perhaps in title). I also miss references to randomness to explain the value of control charts in the abstract (especially when disaggregate data is used, i.e. fewer observations). 

Response: Regarding disaggregation the abstract now refers to use of disaggregated data. The Discussion and Conclusion also describe the importance of disaggregation. Regarding randomness, the paper now includes several mentions of how control charts distinguish noise (randomness) from signal. This includes the third paragraph of the Introduction, the second paragraph in Statistical Process Control within Methods, and the first and fourth paragraph of the Discussion. 

Also in the introduction, the focus is on “displays of Covid-19 data”. The problem with over-aggregation is mentioned (impairing the ability of decision-makers) but I think this could be presented as a (first) separate issue.

Response: We agree with this helpful comment. We edited the abstract so that the first sentence of Methods refers to the use of control charts for studying variation and disaggregating data. We address over-aggregation in the second paragraph of the Introduction. We refer to several common features of publicly available COVID-19 data in this paragraph, including over-aggregation.

On page 4 it is argued that control charts “are less burdensome” and on page 12 you say that “the analysis show that control charts are easily interpretable” and “have a visual simplicity”. First of all you have not analyzed if control charts are easily interpretable, at least not from a (real) decision-maker point of view. There is no data based on decision-makers views. The views expressed are rather arguments usually put forward by scholars, proposing control charts, funnel plots and similar displays to identify special cause variation. 

Response: We appreciate this comment and agree that it is helpful to temper some of these statements in the narrative. We have reduced the discussion of control chart value. Assertations about their interpretability and visual simplicity cite existing literature. We deleted the sentence that stated that control charts can be less burdensome than some other commonly used methods analytic methods. We agree that this point was not justified in the narrative and believe that it would district from the paper to weigh different methods alongside control charts. We edited one sentence so that it states that control charts can be more interpretable. Additionally, the narrative now states that the Los Angeles County Department of Public Health is interested in and beginning to use control charts to assist outbreak management, which is evidence that the control charts are interpretable and add value to the other analytical methods that this local public health authority uses. Several co-authors hold positions within this public health department so we believe this provides some evidence for our assertations about the potential value of these charts. 

If think (and this is supported by experiments) that real decision-makers need at least initial support to fully understand control charts. Support becomes even more important when decision-making is distributed across local authorities (with more limited access to “experts”). Without initial and continued support (training seminars/webbinars, on-line help etc) it would probably be difficult to develop this into “a community engagement tool”. It would also take time - the Covid pandemic require local actions here and now! In summary, I think the article should take the need for training and support involving local decision-makers more seriously. Continued support could also include sharing of local experiences and local knowledge. This could support engaged discussions and improved decisions further.

Response: We appreciate this comment and concur that public health authorities will need support and resources to adopt control chart methods. We do believe that this is possible with a reasonable amount of support, based on our experience introducing this method into the Los Angeles County Department of Public Health and also based on the experience of several co-authors in teaching these methods to state and county public health authorities. This work is cited in the references (Finnerty et al. 2019 and Davis et al. 2016). In response to this helpful and critical point from the reviewer, we have added a final paragraph to the Discussion, as follows:

“As with all significant changes to data reporting and analytics, adopting SPC methods will require time and resource investment. LAC DPH’s interest in using control charts to help inform outbreak management followed a series of virtual workshops on the method that a university partner offered to public health personnel. Health departments will need to create these displays and prepare their workforce to use them effectively, ideally as real-time learning tools. For optimal impact, public health personnel will need to assist community stakeholders to interpret and use the charts to meet their specific needs.”

Additionally, the final sentence of the paper states that “Use of control charts in public health will require an appreciation of their value and investing time and resources into their creation and use. Progress in this area for health authorities and their partnered community stakeholders has been made for topics such as infant mortality through learning collaboratives of state and local health departments [14] and in training programs for local health authorities [30].” 

Reviewer #2: 

1) Methods. In the first paragraph of the methods, the authors describe the characteristics of the control charts that they used in this analysis. However, these specific characteristics do not apply to all control charts relevant to this type of analysis, and I think this introductory paragraph to control charts should emphasize the wide variation in chart characteristics. Perhaps this paragraph would also be better suited for the introduction rather than the methods?

For example, some control charts estimate the centerline from the mean of past data, but other control charts use an expected external baseline rate or other calculations to estimate the baseline. Furthermore, “freezing” a static baseline is one strategy, but other control charts use a dynamic rolling baseline that updates over time. Finally, defining control limits at +/- 3 standard deviations is a common convention but is not required.

Response: We appreciate this comment and now include a paragraph about control charts in the introduction. We also added this referenced sentence to that paragraph in the Introduction to describe common types of control charts: “Control charts can be used for multiple common types of data distributions including classification (binomial) P charts, continuous normal distribution (X charts) and individuals (I charts), and count (Poisson) C charts [12].” Additionally, we made a significant change in the control chart method in this paper. We use the new hybrid control chart developed for the pandemic by Perla et al. that was published in 2020. We describe the specific ways that midlines are calculated and that special cause is identified for this hybrid chart. We believe that this is responsive to the reviewer’s appropriate comment that there are some variations in how initial baseline rates are constructed; rather than describe these variations, we believe it is easiest for readers if we describe the method used in this paper and provide references to papers and texts (e.g., Provost and Murray, The Healthcare Data Guide) that describe these variations in detail. 

2) Methods – control chart analysis. Exponential outbreak growth was quite common in many locations at the onset of the COVID-19 pandemic. The C-charts in this study estimated a centerline from the first 20 observations. However, as expected for a respiratory virus outbreak, exponential growth often occurred within the first 20 days/data points. Does use of a centerline estimated from the first 20 days of data decrease the ability of these charts to demonstrate timely detection of exponential growth within the first 20 days? Does the 20-day baseline need to be separately analyzed for exponential growth before being used as an “in control” baseline for a process assumed not yet to demonstrate exponential growth? I.e., would the baseline be more effective if it were proven to be “in control” before being used to monitor for special cause variation?

Response: We appreciate this comment and as noted in the response to Comment #1 of this reviewer, the paper now describes the published methodology of Perla et al. to create hybrid C and I charts for this study. We briefly describe the rationale for this method and also provide references to the details of this new hybrid chart. The method that we are using is consistent with what the reviewer recommends; that is, the method does establish a centerline of the data that is “in control” while monitoring is underway to identify special cause variation. As noted in the method description, there is a minimum of 8 points to establish an initial baseline for a non-exponential epoch and a minimum of 21 points to establish an initial baseline for an exponential epoch. We believe that this is responsive to the reviewer

3) Methods/Figure 1.

a) Charts c/d/e/f were deemed not to exhibit exponential growth, but many data points ultimately exceeded the apparent upper control limits for these charts. How were these data indicating special cause variation and possible transition to an exponential growth phase determined not to represent exponential growth?

Response: We appreciate that these charts were difficult to interpret. We have replaced all of the charts with the hybrid control chart that Perla et al. developed in 2020. The concern raised by the reviewer is no longer relevant as these charts follow the rule for special cause that Perla et al. established, which is two sequential signs of special cause (versus the standard single sign of special cause). The rationale for this rule is provided in the Methods, and we provide references for the rule. We also note in the Methods that typically a control chart would be monitored for signals of special cause in real time by a person or team that is very familiar with system of focus. In this case, we are generating charts and using special cause rules that follow from those established by Perla et al. for this hybrid chart. 

b) Chart g is labeled as both a hybrid chart and a C-chart – could the authors please clarify this unique feature of the Lancaster control chart?

Response: We replaced all of the charts in Figure 1 with the hybrid chart to address this comment and update the paper. Figure 1 and Figure 2 include the hybrid chart for each area that is displayed. 

c) The Figure 1 legend states that county C-charts used a centerline estimated from the midpoint of all observations, but the methods state that C-chart centerlines were estimated from the first 20 data points. Could the authors please reconcile this discrepancy? Also, does “all observations” refer to all observations up to the current observation or all observations for the entire study? I am asking this question because if observations from May were used to calculate the centerline for April, this method of using a global baseline would not be useful for prospective surveillance.

Response: The method that we are using for the hybrid control chart reconciles the discrepancy that the reviewer identified. All of the charts use the same rules for establishing the initial midline. We heavily edited the Methods section to clarify how the hybrid charts are created, and the confusing language that the reviewer comments on (regarding “all observations”) no longer applies as we have deleted that description. 

d) Could the authors please label Figure 1? Perhaps a single label for all panels would work well, but the “daily counts,” centerline, UCL, and LCL should be labeled, as in Figure 2.

Response: We appreciate this comment and have added a legend to Figure 1. Both Figure 1 and Figure 2 have legends that list the daily cases, midline, UCL, and LCL. 

4) Methods – control chart analysis. The authors defined the end of the exponential phase as when a data point is below the lower limit – when this happens, should an I-Chart be converted back to a C-chart to monitor for re-entry into an exponential phase? If not, it seems that ongoing use of the exponential centerline and control limits is not very useful.

Response: We have rewritten the Methods to explain how special cause is handled in the hybrid control chart. The paper now states that there are four possible epochs (pre-exponential growth represented by a C-chart, exponential growth (an individuals (I) chart), post-exponential growth (a flat trajectory or exponential decline that is represented by an I chart), and stability after descent (C-chart). Additionally, there are phases within an epoch so that the midline can change but the form of the control chart (C, or I) continues. We believe that this method and explanation addresses the reviewer’s comment, and additionally, we have deleted the original control charts that showed the problematic pattern that the reviewer is referring to. 

5) Methods – control chart analysis. My comment here is a generalization of comment #4 above. At the bottom of page 6, the authors state that the centerline and control limits remained constant over time, except when an initial transition to epidemic growth was noted. I think this strategy is a major limitation of this analysis that should at least be discussed in more detail or, ideally, addressed by showing how centerlines and control limits could be adjusted overtime as proof of concept for using these charts for prospective surveillance. For example, in Figure 1h, the majority of datapoints from mid-May onwards are above the apparent UCL, and every data point over this broad timeframe is above the centerline. Clearly, a mechanism is needed to update the centerline and control limits over time for COVID-19 monitoring in order for this chart type to remain useful outside of monitoring for exponential growth during the early stage of the outbreak.

Response: We appreciate these insightful comments and agree that the method was not fully described. As noted in responses to the other methodological questions by this reviewer, the revised paper uses the hybrid chart published by Perla et al. and has specific rules for transitioning between epochs and between phases within epochs. That addresses the reviewer’s question about how centerlines and control limits are adjusted over time. We also cite the methodology paper that describes the approach in detail. 

6) Results – page 14, just prior to conclusions. The authors state that “the purpose was to draw conclusions about causal events,” but I believe this represents a typo and was intended to state that “the purpose was NOT to draw conclusions…”

Response: The reviewer is correct that this was a typographical error. We have corrected the sentence accordingly. 

7) Overall comment – did the authors explore other chart types and chart characteristics for this analysis? How would other charts perform in certain scenarios for monitoring COVID-19 trends? For example, did the authors experiment with different centerline definitions, out of control data point definitions, control limits, etc.? Per my comment above, would other charts provide more useful monitoring after exponential growth has been confirmed or for locations that continue to see variation in case rates without exponential growth?

Response: We appreciate this thoughtful question. As noted in our response to Comment 1 from Reviewer 1, we have added a listing of commonly used control charts, and the Methods section states that this paper uses the two types of control charts that are appropriate for the data distribution underlying counts of COVID-19 cases (C charts or I charts). We believe that the new hybrid control chart methodology addresses the helpful points that the reviewer is making, in its use of C and I control charts within the same display. The Methods section now explains that the data may signal a change of phases within a C or I chart or may signal a change in epoch, which means a shift from a C to an I chart or a shift from an I to a C chart. Additionally, we provide a rationale for the rule of identifying a signal of a phase or epoch shift when there have been 2 sequential observations that indicate special cause. To avoid adding information that readers might consider to be extraneous, we do not go into detail about how other kinds of control charts could be used for understanding COVID-19 data, such as charts that are appropriate for rare events or that are used for binomial distributions.

---

## [Editor Report · Decision Letter 1]

22 Feb 2021

PONE-D-20-24209R1

Using Control Charts to Understand Community Variation in COVID-19

PLOS ONE

Dear Dr. Inkelas,

Thank you for submitting your manuscript to PLOS ONE. After careful consideration, we feel that it has merit but does not fully meet PLOS ONE’s publication criteria as it currently stands. Therefore, we invite you to submit a revised version of the manuscript that addresses the points raised during the review process.

I participated as a reviewer for the initial evaluation of this manuscript and believe that you have made important improvements to the manuscript in responding to reviewer feedback. If you can make a few additional minor changes to improve readability, I anticipate that your manuscript will fully meet PLOS ONE's publication criteria.

The following three areas require revision or further clarification:

1. Results, page 12. The results state that household crowding at Westlake occurred in 45% of households, but the revised Table 1 lists 45% for Westlake household crowding. The results also describe Westlake as having a large household size with median size of 4.0; however, Table 1 gives median household size of 3.0 for Westlake, which is the second smallest household size among the five cities/neighborhoods studied. Could you please clarify this apparent discrepancy and ensure that all table/manuscript statistics are accurate?

2. Results, page 12. I recommend removing the last paragraph that mentions publicly available stratification of counts. This comment is useful but is better suited for the discussion section. This comment could be folded into the related discussion of general public vs. congregate living populations on page 14.

3. Discussion/Conclusions sections. I appreciate the revised and improved discussion and conclusions sections. Additional copyediting of these sections by the authors would improve readability and impact. While these changes would be stylistic and overall minor, I think editing by the authors at this stage is important because the journal does not provide copyediting. Here are some examples of the type of copyediting that may improve readability of these sections:

a) Page 13, line 3. Could consider: “...tool for daily learning and action that could benefit public health.”

b) Page 13, last sentence of paragraph 1. Could consider: “Use of control charts could reduce overreaction to expected random variation and encourage immediate action when…”

c) Page 14, second paragraph. Could consider: “Data alone do not tell the full story.”

d) Page 17. Could consider: “Use of control charts in public health will require an appreciation of their value and investment of time and resources into their creation and use.”

We look forward to receiving your revised manuscript.

Kind regards,

Arthur Wakefield Baker

Academic Editor

PLOS ONE

---

## [Author Response · Author response to Decision Letter 1]

24 Feb 2021

Thank you for the helpful review. We addressed each comment in the revised narrative. Our responses to the reviews are provided below.

1. Results, page 12. The results state that household crowding at Westlake occurred in 45% of households, but the revised Table 1 lists 45% for Westlake household crowding. The results also describe Westlake as having a large household size with median size of 4.0; however, Table 1 gives median household size of 3.0 for Westlake, which is the second smallest household size among the five cities/neighborhoods studied. Could you please clarify this apparent discrepancy and ensure that all table/manuscript statistics are accurate?

Response: We corrected a typographical error in the narrative that resulted in a discrepancy between the narrative and Table 1. We deleted a sentence that had some redundant information. We reviewed the data in the narrative, and in Table 1, and all other data are correct. 

2. Results, page 12. I recommend removing the last paragraph that mentions publicly available stratification of counts. This comment is useful but is better suited for the discussion section. This comment could be folded into the related discussion of general public vs. congregate living populations on page 14.

Response: As suggested, we deleted this last paragraph from Results and added this point into the Discussion (Page 15) as follows: “Local jurisdictions may need this differentiation to target strategies – for example, if most cases are in congregate facilities.”

3. Discussion/Conclusions sections. I appreciate the revised and improved discussion and conclusions sections. Additional copyediting of these sections by the authors would improve readability and impact. While these changes would be stylistic and overall minor, I think editing by the authors at this stage is important because the journal does not provide copyediting. 

Response: We appreciate the suggestion of additional copy editing. We combined two paragraphs in the Discussion and made additional line edits to improve the clarity of this section. The changes are tracked in the revised narrative. Additionally, we addressed the specific copyediting suggestions (please see below). 

Here are some examples of the type of copyediting that may improve readability of these sections:

a) Page 13, line 3. Could consider: “...tool for daily learning and action that could benefit public health.”

Response: We made this edit in the narrative. 

b) Page 13, last sentence of paragraph 1. Could consider: “Use of control charts could reduce overreaction to expected random variation and encourage immediate action when…”

Response: We made this change. 

c) Page 14, second paragraph. Could consider: “Data alone do not tell the full story.”

Response: We deleted this sentence and instead edited the next sentence to read: “Chart annotation makes it possible to…”

d) Page 17. Could consider: “Use of control charts in public health will require an appreciation of their value and investment of time and resources into their creation and use.”

Response: We made this edit in the revised narrative.

---

## [Editor Report · Decision Letter 2]

1 Mar 2021

Using Control Charts to Understand Community Variation in COVID-19

PONE-D-20-24209R2

Dear Dr. Inkelas,

We’re pleased to inform you that your manuscript has been judged scientifically suitable for publication and will be formally accepted for publication once it meets all outstanding technical requirements.

Kind regards,

Arthur Wakefield Baker

Guest Editor

PLOS ONE

---

## [Editor Report · Acceptance letter]

19 Apr 2021

PONE-D-20-24209R2 

Using Control Charts to Understand Community Variation in COVID-19 

Dear Dr. Inkelas:

I'm pleased to inform you that your manuscript has been deemed suitable for publication in PLOS ONE. Congratulations! Your manuscript is now with our production department. 

Kind regards, 

on behalf of

Dr. Arthur Wakefield Baker 

Guest Editor

PLOS ONE